# The Experience of the Transition from a Student Nurse to a Registered Nurse of Kuwaiti Newly Graduated Registered Nurses: A Qualitative Study

**DOI:** 10.3390/healthcare10101856

**Published:** 2022-09-23

**Authors:** Fatmah Kreedi, Michael Brown, Lynne Marsh

**Affiliations:** 1Medical Biology Centre, School of Nursing and Midwifery, Queen’s University Belfast, 97 Lisburn Rd, Belfast BT9 7BL, UK; 2Public Authority of the Disabled, Kuwait City 34R5+25Q 212, Kuwait

**Keywords:** newly graduated registered nurses, turnover, job satisfaction, support, retention, transition, social image

## Abstract

Background: The experience of the transition from a student nurse to a registered nurse is a challenging period for newly graduated registered nurses. Aim: To explore newly graduated registered nurses’ experiences of transition from student to registered nurse in clinical practice. Design: A qualitative approach using semi-structured interviews conducted with 12 Kuwaiti newly graduated registered nurses. Findings: The findings generated three themes: nursing support; education preparation; and psychological wellbeing. Discussion and conclusion: This study is the first in Kuwait aiming to understand Kuwaiti national newly graduated registered nurses’ transition experiences from student nurses to registered nurses in clinical practice. While the study revealed that newly graduated registered nurses received limited organisational support, the nursing policymakers in health care organisations and nursing education in Kuwait need to develop plans to improve newly graduated registered nurses’ knowledge, skills and confidence and align them with the roles and realities of actual nursing practice, to improve retention. There is a need to change the societal image of nursing in Kuwait by highlighting the importance of the nursing profession within the health care delivery. The study recommends further research on newly graduated registered nurses’ transition experiences into their new nursing roles to identify the factors behind their decision to stay or to leave, as this could offer possible solutions to address newly graduated registered nurses’ retention in the future.

## 1. Introduction

Registered nurses make up the largest element of the healthcare workforce and make major contributions to patient care for children, adults and older people [1]. The World Health Organisation (WHO) estimates a global shortage of over 7.6 million nurses by 2030, with implications for healthcare delivery [2]. Additionally, there is growing recognition of the impact and implications of the high turnover rate of newly graduate registered nurses (NGRNs) at the outset of their professional career. Internationally, for example in South Korea, NGRNs turnover was 18% within the first year as a registered nurse, rising to 47% within the first three years [3]. In the United States of America (USA), registered nurse turnover levels of 13% within the first year were identified in an integrative review of job-related stress experienced by newly registered nurses [4,5]. The findings evidenced that some NGRNs experienced high stress levels due to heavy workloads and a perceived lack of professional nursing competencies. In a qualitative study involving 15 NGRNs from Finland, the findings highlighted poor nursing practice environments, limited support, bullying and workload and work stress as the main contributory factors for NGRN turnover [6].

NGRNs turnover negatively affects healthcare delivery and patients care. Evidence points to the consequences of NGRNs turnover resulting in poor health-related outcomes for patients, including reduced patient satisfaction, increased patient falls, increased pressure ulcers, medication errors, diagnosis and treatment errors and increased length of admission [7,8]. The situation is further compounded by the growing financial burden of increasing healthcare demands as populations age [9]. A qualitative study conducted in Kuwait to estimate the future demand for nurses in the country between the years 2007 and 2020 found that inadequate nurse staffing, heavy workloads, the use of overtime, limited support from colleagues and nurse managers and inadequate salaries were among the factors identified as contributing to nursing turnover [10]. In Kuwait, the nurse’s role is viewed as passive and lacking in decision-making power; moreover, nurses are perceived as patients’ servants, and doctors’ assistants [11]. These negative images of nursing not only appear to exist in Kuwait, but are also an issue that exists in the surrounding Gulf area and other countries. Findings from a qualitative case study conducted in Oman aimed to explore NGRNs’ transition experience from one university in the Sultanate of Oman with a sample of 52 of NGRNs, student nurses, nursing instructors, head nurses, preceptors and managers [12]. The study evidenced that nursing is not an attractive choice for Omani students to study and pursue as a future career.

The Kuwaiti population is growing with increasing older people, many with long-term health conditions [13,14,15]. As with the global situation, Kuwait has a shortage of registered nurses, and has relied on non-Kuwaiti national health professionals to support the expanding health care system [9]. In 2017, the nursing workforce comprised 23,015, with 21,913 non-Kuwaiti and 1102 Kuwaiti national registered nurses [15]. The situation is further compounded by the shortage of Kuwaiti national NGRNs with a rise in resignations [13]. Currently, in Kuwait, pre-registration education in nursing is provided by Nursing Institutes and Colleges of Nursing, both are under the umbrella of the Public Authority for Applied Education and Training (PAAET). Nursing Institute, accept students who had completed 9 years of general education [16]. The Nursing Institute body awarded nursing certificates. Students were required to complete 12 years of general education before entering the College of Nursing [17]. Currently, the College of Nursing offers the following academic programmes: 2.5 years (5 semesters) Associate Degree in Nursing (ADN), and Bachelor of Science in Nursing (BSN) degree with three options; 4 years (8 semesters) Bachelor of Science in Nursing (BSN-generic), 2 years (4 semesters) Bachelor of Science in Nursing (BSN-post-basic), and 4 years (8 semesters) Bachelor of Science in Nursing (BSN-School Health Nursing) [17].

The College of Nursing employs a variety of frameworks to direct academic progression and to ensure that student learning objectives are successfully and efficiently met [17]. The frameworks and guidelines used include the Accreditation Commission for Education and Nursing, Adult Learning Theory, Bloom’s Taxonomy of Learning, International Council for Nurses (ICN), and American Nurses Association (ANA). Before starting practical training, the students need to study the Biomedical courses such as Anatomy and Physiology, Organic and Biochemistry and Applied Nutrition. However, practical training for students in different settings identified according to the different nursing semesters they are taking such as foundations of nursing, adult health Nursing, community, mental health nursing and Intensive Care Practice. The time spent in the clinical practice for student nurse is six hours, from 7 am to 1 pm three days per a week a combined with their Nurse Educators, while the College of Nursing delivered the nursing curriculum in English, the Nursing Institute delivered the nursing curriculum in Arabic.

### 1.1. Significant of the Study

It is widely recognised globally that the transition from student nurse to registered nurse is a challenging time in the career of NGRNs [18,19]. During this period, NGRNs are confronted with a number of challenges, such as work overload and unsupportive work environments that can affect their professional life and transition experiences [20]. The findings were similar across studies with many linkages to NGRN turnover, namely unsupportive work environments, work stress, fear of inadequacy, significant gaps between theoretical and practical knowledge, frustration, and excessive workloads [12,19]. Similarly, studies from Canada [21], the USA [22], and Singapore [23] acknowledged that NGRNs found this transition period very stressful. Oneal et al. (2019) conducted a qualitative study in Washington, USA, to explore the transition of NGRNs (n = 34) into clinical practice concerning the framework of total work, safety, and health, using semi-structured interviews and focus groups interviews [18]. The study suggests that factors both at home and work can affect NGRNs’ wellbeing and lead to burnout and turnover [18].

When set within the wider global context of nursing shortages today and in the coming decades, and the challenges and needs of NGRNs, there is a pressing need to more fully understand these issues in the Kuwaiti context and to identify possible solutions [18,19]. However, to date, this has not been reflected in either the design or reporting of the research studies conducted in Kuwait investigating the experience of transition from student nurse to registered nurse of Kuwaiti national NGRNs in clinical practice. Therefore, understanding NGRNs’ experiences of transition at the beginning of their professional careers is a matter of global concern, specifically regarding the Kuwaiti context in order to help meet NGRNs’ needs and assist in retaining them in the profession. Specifically, the findings of this study aim to provide in-depth insights into Kuwaiti national NGRNs experiences, the challenges they encounter and the reason behind their decision to stay or leave the nursing profession. These issues need to be addressed to ensure that the nursing workforce in Kuwait is developed and adequately prepared to meet future health service demands and contribute to the delivery of safe, effective and person-centred patient care. Moreover, a Kuwait- and Gulf-wide research network should be established to enable cross-state collaborations and comparisons in relation to nursing workforce needs and developments.

#### 1.1.1. Research Aim

This study aimed to explore the views and experiences of the transition from student nurse to registered nurse of national Kuwaiti NGRNs in their first post in clinical practice

#### 1.1.2. Research Questions

What are the views and experiences of national Kuwaiti NGRNs, regarding the transition experience from nurse student to registered nurse in Kuwait?

The three sub-questions following from the main question:

What supports are required by NGRNs to facilitate their transition to practice?

What are the barriers that may hinder NGRNs may transition to practice?

What are the possible solutions to improve the experience of NGRNs when they transition to practice?

## 2. Materials and Methods

### 2.1. Research Design

This study was based on qualitative semi-structured interviews comprising 12 Kuwaiti national NGRNs working in government hospitals and Primary Health Care Centres from five health regions in Kuwait. The data were analysed using thematic analysis framework [24]. There is a need to understand the Kuwaiti national NGRNs’ transition experiences from student nurse to registered nurse to clinical practice, in order to provide effective support and develop appropriate strategies for retaining this population and help prevent turnover, and to ensure adequate patient quality care. Qualitative research was identified as the most effective method to answer the research questions regarding individuals’ experiences and impact on their lives [25]. Exploratory research designs are used where there is a limited established evidence-base and understanding in connection to a specific scenario, group, activity, or process to be examined [26]. Exploratory research is considered as a flexible design used to discover new deep insights, comprehend the phenomena, ask questions, and appraise the subject from a fresh perspective [27,28]. Given the nature and intended focus of the proposed study, an exploratory research approach was deemed the most appropriate [29]. There is a limited and evolving body of research evidence regarding the transition experience of national NGRNs in Kuwait. An earlier study conducted by Alotaibi (2008), used a qualitative research approach to study nursing turnover in Kuwait but did not focus on national NGRNs [30]. Furthermore, recently, Alnuqaidan et al. (2021) explored multinational NGRNs’ transitional shock through assessing occupational stress and coping mechanisms in Kuwaiti health services [31]. Alnuqaidan et al. (2021) studied multinational NGRNs but explored only one area of transition issues, namely transitional shock and the relationship of stress and coping mechanisms whereas this current study focused solely on Kuwaiti national NGRNs and identified a variety of issues related to their transition experience, including transition shock, stress and coping strategies [31]. As no previous qualitative study has been conducted on the views and experiences of national NGRNs during the transition period from student nurse to registered nurse in Kuwait from the perspectives of national NGRNs, it was important to explore and understand this issue and developments required using a qualitative research approach. Therefore, exploratory qualitative research design was the most appropriate approach for this study to achieve the study aim and answer research questions. 

### 2.2. Sampling and Recruitment

A purposive sampling strategy was adopted to identify the participants who met the inclusion criteria [32]. Local gatekeepers from the Staff Development Unit in each region were identified and disseminated details about the study to potential participants and the contact details of the researcher, and the inclusion and exclusion criteria. To be eligible to participate, the NGRNs had to have commenced their nursing role in clinical areas in one of the five regions between January 2016 and October 2017, Kuwaiti nationals, undertaken their nurse education in Kuwait and currently employed in government health services in Kuwait. The exclusion criteria were employees not NGRNs (Senior nurses, Head Nurses), NGRNs not working in government health services, and non-Kuwaiti nationals

Twelve interviews (n = 12) were conducted with NGRNs. Ethical approval was obtained from the Ministry of Health in Kuwait, with all ethical processes followed throughout the study. The ethical framework proposed by Beauchamp and Childress (2001) was selected for use in this study. This ethical framework consists of four principles: autonomy, beneficence, non-maleficence, and justice. All participants provided informed consent and completed a consent form.

### 2.3. Ensuring Rigour

Ensuring rigour with the aim of ensuring the quality of research findings requires clarity in the research process, and how rigour was achieved throughout. Lincoln and Guba (1985) identified four criteria for evaluating trustworthiness in qualitative research: credibility, dependability, transferability, and confirmability, all of which were applied by the researcher in the current study [33].

Several steps were undertaken in the current study to ensure credibility by the researchers. Firstly, the semi-structured interviews were guided by the interview schedule and were fully conversant with reflective questioning. Secondly, all interviews were audio recorded and transcribed verbatim. Dependability was ensured by the researchers following the guidance provided by the ethics committee and ethical principles throughout the research process, conducting member checking, ensuring that each participant in the study examined verbatim transcripts and verification of themes identified. The researchers also provided a detailed audit trail for each stage of the research process, so that the study could be replicated in the future. In this study, the researchers achieved transferability by providing a detailed description of the research process and results, together with the research method, sampling, and the use of selection, inclusion and exclusion criteria, data collection, and the methods of analysis. To evidence confirmability in this study, the researchers presented a detailed methodology outline as well as details of the audit trail to enable readers to confirm that the researchers’ analysis of the results and interpretation of the data were accurate.

### 2.4. Data Collection Methods

The interview schedule was developed based on the research literature and aim of the study, with two pilot interviews completed [34]. The length of each interview was between 20 to 55 min, with an average of 37 min. Data saturation was achieved, as no new themes or additional information emerged from the data collected [35].

### 2.5. Data Analysis

All the semi structured interviews were conducted in Arabic, digitally recorded using a voice recorder and transcribed verbatim from Arabic to English. All identifying information in the transcripts was deleted, and any references to participants’ names were substituted with pseudonyms. The transcripts were also checked and reviewed by a second individual who was a language expert, to confirm their accuracy. The data analysis was guided by thematic analysis framework [18]. Confirmability of the findings was achieved by developing an audit trail that included evidence of the interview data, data reduction methods, process notes, reflexive notes and an outline of the researcher’s intentions and dispositions throughout the study [33].

## 3. Results

### 3.1. Participant Demographics

All 12 participants were NGRNs, female, aged between 20 and 34 years, of whom seven were married. Six had graduated from the Nursing Institute with a Certificate in Nursing, five had graduated from the College of Nursing with an Associate Degree in Nursing and the remaining participant had graduated from the College of Nursing with a Bachelor of Nursing degree. Five participants were employed in Primary Health Care Centres, two in paediatric wards and the remaining in operating theatres, outpatient clinics, surgical wards, maternity wards or esophagogastroduodenoscopy department (EGD) and all had completed their orientation programmes (see Table 1). Data analysis resulted in three themes: nursing support; education preparation; and psychological wellbeing.

### 3.2. Theme 1: Nursing and Family Support

The most common theme identified that influenced the retention of NGRNs was associated with the support provided by their health care organisation, family and friends. Participants suggested that better organisational support could be provided by the Ministry of Health, nursing administration and colleagues in the clinical environments. Specifically, low salary and lack of support by their health care organisations were an important reason why Kuwaiti nationals were reluctant to pursue a career in nursing:

“…there is a lack of support for nurses. They do not value nurses here. I feel frustrated, [pauses, looks down at floor] …nothing, no support, no value, and no good salary. This is an important reason why nurses leave.”(Safia, NGRN)

“All we need is someone to hear us, to hear our problems.”(Hafsa, NGRN)

Five participants were of the view that they had not experienced a supportive work environment in their first post as they did not feel welcomed and included by colleagues. The belief was that some senior nurses grouped together according to. nationality, supporting each other, thereby failing to support Kuwaiti NGRN, which made their transition considerably more difficult.

“Some nurses become groups and who are the same nationality helping each other and not helping other new nurses.”(Asma, NGRN)

However, six participants reported that the support they received in the work environment had a significant positive impact on their satisfaction and helped retain them in nursing:

“The most important factor that made me feel satisfied was the support from the work environment and staff cooperation.”(Aisha, NGRN)

One of the important forms of administration support identified by the participants was linked to the hospital orientation programme, or lack thereof. NGRNs overall reported that the orientation period was a positive experience, helping them to become familiar with new clinical settings, hospital regulations, policies and gain self-confidence and by providing support during the transition period.

“The orientation period helped [support] me very much in making me more confident. The orientation was 3 months long in a different department over 9 months. That made me feel that I am qualified enough to start my career”.(Sheikha, NGRN)

Another organisational support strategy that could be offered to NGRNs to help with the transition was access to continuing education, to improve their knowledge, personal skills, and social and clinical competence. Five participants reported that continuing education was important for them to update their nursing skills and expand knowledge further following graduation:

“For our job there is always something new. Therefore, we need to educate ourselves more and expand our knowledge wider than what they teach us in Nursing Schools.” (Hafsa, NGRN)

Negative public attitudes were another significant issue reported by NGRNs during their experiences of transition, leading them to consider leaving the profession. Seven NGRN participants believed that the social image of nurses in Kuwait was a significant barrier that led them to question their career choice and remaining in the profession:

“The nursing profession has a poor image within Kuwaiti society. I do not think that it is a role that is valued and respected.”(Aisha, NGRN)

Four participants highlighted that support received from family and friends during their transition from student nurse to registered nurse was invaluable by encouraging them to remain in the profession:

“My mother was supportive; she loves the profession [being a nurse] and this was positive for me and helped to encourage me to stay as a nurse.”(Sarah, NGRN)

### 3.3. Theme 2: Education Preparation

Education preparation sets out how the experiences of undergraduate student nurses were viewed by all study participants by examining the perceived role of these experiences on the transition of NGRNs to practice. Nine NGRNs reported that the curriculum theory content in Nursing Schools provided suitable preparation for professional practice, leading to professional registration. However, practice-training experiences were identified by seven NGRNs as an important factor enabling students to apply theoretical knowledge and obtain a greater and broader understanding of the study content to begin their professional careers with more confidence:

“The theoretical lectures qualified me to be a nurse. Now in my work, I can see everything that I learnt. I find nothing strange; it was all helpful in preparing me for my future in nursing.”(Norah, NGRN)

“It [practical training] helped me to practice the hospital routine and we learnt everything that happened in the wards. Everywhere I trained helped me in removing the fear that happens when anybody starts the first-time job. Yes, I felt that field training was useful.”(Zainab, NGRN)

However, the teaching of those practical skills was time-limited and often insufficient. NGRNs indicated that they did not have adequate learning opportunities within practice settings along with the training time allocated was short due to curriculum time constraints:

“I think that students need more practice time to prepare them, especially in the ICU and CCU, they did not train us enough on it.”(Hafsa, NGRN)

English is the dominant language within all health care settings in Kuwait. While NGRN participants who graduated from the College of Nursing did not report any difficulties in verbal and written communication within practice settings, as the College of Nursing deliver the nursing curriculum in English. The finding indicated that four NGRNs attributed their limited command of English to the method of teaching within the Nursing Institute:

“…the theoretical lectures in the College of Nursing have helped a lot. They helped me to develop my English, and this helped to build my confidence in practice.”(Sheikha, NGRN)

“While our studies were in Arabic in the Nursing Institute, we were asked to deal in English at work. So, it is difficult for the student who does not know English well enough. This is a definite disadvantage.”(Hafsa, NGRN)

English language weaknesses were found to limit the proficiency of NGRNs, which impacted on their ability to confidently provide nursing care within practice settings as part of the wider health care team.

### 3.4. Theme 3: Psychological Wellbeing

The transition from student to registered nurse was found to have a significant impact on NGRNs in the psychological, physical, professional and social functioning and were viewed as interrelated. Stress was found to have a negative impact on the transition experience of Kuwaiti NGRNs. The participating NGRN expressed feelings of stress in different ways: feeling shocked, frustrated and isolated, experiencing fear of new situations, feeling undervalued, and overworked. Five NGRN participants were shocked by the reality that their practice placement and expectations differed from what they had studied and prepared for, and that theoretical knowledge was difficult to apply in real-life clinical practice settings:

“We were shocked by the reality because here no one sticks to the steps we were trained to do. Even if you try to apply what you were taught, other nurses get very nervous and angry because they want to finish the job as quickly as possible.”(Mariam, NGRN)

Feelings of shock typified the frustration and isolation experienced by NGRNs when they perceived the gap between what they were taught, what they expected from the nursing role, and what they observed in actual practice when senior staff did not apply the care principles they had learnt. Two NGRNs revealed that when they commenced work as registered nurses, they recognised significant gaps in their knowledge and skills acquisition, and realised that they would require further ongoing training and practice development to become competent practitioners:

“At the beginning, I thought I was prepared but I realised that I have to do the actual work first. It is different when I was just watching and supervising, and actually doing the work.”(Hafsa, NGRN)

Six NGRNs reported experiencing fear and stress when they first became registered nurses, which was further exacerbated by their inexperience. Three of these participants reported that the fear was associated with a fear of making mistakes, and facing stressful situations for the first time:

“I feel stressed because of fear of making a medical mistake, afraid to do something and then worsen the patient’s condition and will be held accountable.”(Reem, NGRN)

Workload demands were also reported by NGRNs as an additional source of stress. NGRNs reported that heavy workloads increased their responsibilities due to nursing shortages and they became overwhelmed. Two NGRNs reported that within practice settings they had experienced increased workloads, which, at the same time, increased competing demands related to the care needs of patients and other responsibilities in their new role:

“The negative things were the shortage of staff and workload. This had a big effect on me in my new nursing role.”(Amina, NGRN)

While all NGRNs perceived that the transition to registered nurse was stressful, some reported using a range of different strategies to cope with the stress they encountered. For example, being positive and motivated was identified as an important coping strategy by participants, which helped them cope with the new work setting. The desire to be a nurse was one of the self-motivators, which encouraged participants to remain in the nursing profession even with the difficulties they experienced.

“Since I was a child, I wanted to become a nurse, I loved the profession.”(Zainab, NGRN)

Indeed, nine NGRNs reported that using coping strategies enhanced their self-confidence and self-development, and the ability to continue on their career trajectory:

“I broke the fear barrier, increased my self-confidence and increased my skills. This has helped me to develop as a registered nurse and I am happy that I did not give up.”(Aisha, NGRN)

Another factor that motivate NGRNs and assist their satisfaction stemmed from appreciation from patients and their families and engaging in direct care and nursing critically ill patients:

“I feel that I am satisfied when I am appreciated by my patient because I feel that I helped him. This is very special and reminds me of why I wanted to become a nurse. It is what nursing is about, helping people.”(Norah, NGRN)

On the other hand, some NGRNs appeared to cope less with the stressors they experienced. Six participants openly expressed the view that they were seriously considering leaving their current nursing positions, due to excessive stressful experiences that occurred almost from the beginning of their professional career:

“… I want to leave nursing because from the beginning I felt I was not developing in the nursing profession. No support, no encouragement.”(Reem, NGRN)

Norah indicated that she wished to undertake further study while emphasising that this would not be an option open to her in nursing, and required a career change:

“I intend to resign and study another major and become a teacher, which will mean leaving nursing after all my training.”(Norah, NGRN)

Safia, however, expressed the desire to seek work opportunities in another country, because of the additional support her friend had received in Canada:

“I am thinking to go to Australia and work there. My friend went to Canada, she found a lot of support and she is very happy there.”(Safia, NGRN)

## 4. Discussion

The current research aimed to explore the views and experiences of the transition from student nurse to registered nurse from the perspective of Kuwaiti NGRNs. Findings from the study evidenced that support from the health care system has a significant impact on NGRN retention. For example, access to a structured orientation programme with designated mentoring is a support that health care organisations can offer to NGRNs to ensure retain them in the health care setting. A consensus is evident across studies on the importance of orientation programmes delivered to NGRNs as one strategic process of mobilising appropriate organisational support [36]. Additionally, findings from this study show that NGRNs’ sense of feeling supported by their families and friends, feeling appreciated by patients and their family members, or of having a positive experience in the workplace were reasons for many to remain in the profession. The current study findings are echoed by Clark and Springer (2012), and Parks et al. (2013), who found that being appreciated by patients and their family members had a significant impact on NGRNs’ self-confidence, job satisfaction, and retention. [20,37].

On the other hand, this study evidenced that NGRNs experienced a number of stressors in their work environments. For example, lack of support from health care organisation such as low salaries, lack of colleagues supports and excessive workload. This finding reflects those of a three-year longitudinal study carried out with 38 NGRNs and 12 Head Nurses in Australia which compared NGRNs’ and Head Nurses’ perceptions regarding the workplace factors that affect NGRNs during their first year of clinical practice [38]. Additionally, low financial remuneration led to the possibility of nurse attrition. This study concurs with a study conducted by compared job satisfaction among nurses in some European countries [39]. The study found that low salary was reported as a major factor associated with job dissatisfaction for nurses [40]. To address these concerns, it was recommended that the Ministry of Health should increase nurses’ salaries to enhance NGRN retention and to increase the attractiveness of the nursing profession as a long-term career option. This is important as the support given to NGRNs impacts significantly on their retention and minimises their decision to leave the nursing profession. Nurse retention has been identified as a significant global concern by the World Health Organisation (2016), since both developed and developing nations confront a nursing workforce deficit [2].

Findings identified that some Kuwaiti NGRNs have weaknesses in English language, notably the Nursing Institute graduates, where the programme is taught in Arabic. This finding is important and relevant to the NGRN role, as the Nursing Institutes provide the curriculum in Arabic, while NGRNs need to communicate in English in clinical practice. As a result, graduates encountered difficulty communicating with non-Arab health care colleagues and patients, and experienced considerable challenges with nursing documentation and report writing, impacting on their new role as registered nurses. These findings concur with a cross-sectional descriptive Saudi study on 40 patients with cardiac conditions, conducted by Almualem et al. (2021), which investigated whether the language barrier increased anxiety for patients admitted with cardiac diseases to the coronary care unit [40]. Most of the staff nurses working in these centres were expatriates who knew little Arabic, the only language used by almost all of the patients. A questionnaire used to assess anxiety levels revealed that patients who were cared for by Arabic-speaking nurses had a lower collective mean for the anxiety domain statements than those who were cared for by non-Arabic-speaking nurses, which could affect the quality-of-care delivery. Therefore, the issue of language competency is an important issue and finding in the current study that needs to be more fully understood given the large number of international nurses who comprise most of the Kuwaiti nursing workforce.

Findings from the current study suggest that although the nursing curriculum in Kuwait is comprehensive and up to date, it was suggested that nursing students within the nursing institutions require further preparation for their future role as registered nurses in clinical practice. This was evidenced by the NGRNs reporting that training opportunities within clinical settings were sometimes inadequate for students. This was due to limited time allocated to education due to competing demands within the curriculum. This is in line with Clipper and Cherry (2015), who also found that while student nurses met the minimum clinical practice hours, practice placement time was often limited in providing direct patient care [41]. Therefore, this study identifies the need to transition to a baccalaureate degree for nurses in Kuwait as a key action arising from the findings of this study. Furthermore, access to learning opportunities can be facilitated to support ongoing education through educational grants or scholarships [42].

In this current study, NGRNs experienced negative public attitudes, which influenced their decision to remain within the nursing profession and made their transition into the new role more challenging. This negative image of nursing appears to exist not only in Kuwait but also in the wider Gulf area as well as other countries. Findings from a study conducted in Oman by Al Awaisi et al. (2015) evidenced that nurses remain stereotyped as servants, with nursing viewed as a profession of low social status involving physical activities that do not require formal qualifications [12].

### 4.1. Male Nurses in Kuwait

A new and important finding arising as a result of the current study is that no male NGRN participated. Historically, nursing in Kuwait has been dominated by females, with male national nurses still very much a minority of the nursing population. Men comprise 9.6% of registered nurses in the USA [43], 11% in Australia [44], and 10.6% in the UK [45]. In contrast, in 2017, national male nurses stood at just 0.5% of registered nurses in Kuwait [13].

There is a dearth of research about male nurses’ experiences internationally and in Kuwait specifically. In Kuwait as elsewhere, nursing has long been promoted as a female profession, and therefore, being a male nurse in the Kuwaiti community is not viewed favourably. The findings from a phenomenological study conducted on Jordanian male nurses by Saleh et al. (2020) found that although male nurses found themselves more independent in decision making than female nurses, they face a number of social constraints in the Arabic community [46]. Given that Arabic societies are similar in terms of their cultural norms, and due to the small numbers of male nurses and lack of local studies in Kuwait, one can say that Kuwaiti society has the same perception of the nursing profession as other Arabic countries, which leads to societal barriers for males.

### 4.2. Implications for Nursing Policy

Health care policymakers need to develop a Kuwaiti-wide long-term strategic plan to promote a positive societal image of nursing as a professional and viable career through community and school campaigns to correct long-standing myths and inaccurate stereotyping of nursing as a profession and to increase number of nurses. Additionally, health care policymakers need to develop a Kuwait-wide long-term strategic plan to promote nursing as a viable career opportunity by increasing their salary and affording them the opportunity of continuing their study and professional development.

### 4.3. Implication for Nursing Practice

There is an opportunity for policymakers in Kuwait to develop and implement a NGRN orientation programme that involves nurses and other healthcare professionals as a means of ensuring that the needs and concerns are recognised and addressed that listen to and responds to the experiences of NGRNs in practice. The ultimate outcome of such a programme is to develop NGRNs’ understanding of organisational policy, processes, organisational mission, vision and values, and thereby assist NGRNs to contribute more effectively in achieving quality care and improving patient safety [47]. Furthermore, there is a need for policymakers to foster a supportive environment at the point of transition to help retain NGRNs in their posts. This could be achieved through flexible duty shifts, and NGRNs feeling recognised, valued, supported, and encouraged through positive feedback provided by Nurse Directors.

### 4.4. Implication for Nursing Education

In Kuwait, policymakers need to develop and implement a long-term strategic plan to upgrade all nurse education to a baccalaureate degree programme, taught in English. In line with international developments. Furthermore, policymakers in Kuwaiti nursing education need to adjust the nursing curriculum to the demands of the health care system, as well as review student nurses’ clinical placements to improve their clinical experience and bring them in line with the realities of the nursing role. Additionally, global policymakers, such as the World Health Organisation and the International Council of Nurses recognise the importance of increasing investment in nursing education as a critical approach for strengthening the nursing workforce and improving health care systems by upgrading entry level nursing education to baccalaureate degree level. By doing so, the opportunity to build capacity within the educational system to adequately prepare more nurses would be created, thereby growing future leaders, educators, and researchers [48]. Furthermore, policymakers should work to establish a nursing college in every region in Kuwait to increase the number of student nurses and NGRNs for the future. Additionally, policymakers and nurse educators need to develop a Kuwait-wide, long-term strategic plan to support the development of access to postgraduate nurse education. Providing this support to NGRNs is critical in reaping the benefits of health care development and supporting both retention and professional development

### 4.5. Implication for Further Research

To date, there is a dearth of nursing research undertaken in Kuwait. To ensure that nursing is recognised as a respected profession in Kuwait, the evidence and knowledge base needs to be specifically developed for the Kuwaiti context and to meet the country’s health care service demands. This study highlighted several areas that need to be addressed regarding nursing policy, practice, and education that need to be studied and explored further to provide Kuwaiti generated solutions. There is an opportunity for this study to be replicated with a larger sample of student nurses, NGRN’s, Nurse Educators, Nursing Directors and Head Nurses across Kuwait who represent a wider range of socio-demographic backgrounds, including male nurses. Furthermore, there is a need for longitudinal research to explore the impact of NGRN programmes on role satisfaction and retention, the impact of nurse recruitment programmes, and changes in societal attitudes towards nursing as a profession. Following on from this is the opportunity to establish and develop a Kuwait- and Gulf-wide research network to enable cross-state collaboration and comparison in relation to nursing workforce needs and developments, including those of NGRNs as they transition into their new nursing roles.

### 4.6. Strengths and Limitations of the Study

Firstly, to date, no research has been conducted in Kuwait exploring national NGRNs’ views and experiences of transition from student nurse to registered nurse. Therefore, this study offers the starting point to informing the on-going development of the nursing profession in Kuwait and contributes to the wider international evidence-base. The researcher in this study was not employed by the Ministry of Health where the data collection took place, nor was there any personal or professional relationship with any of the participants.

The researcher’s data analysis was aided by audit trails that included notes on where and when the interviews were conducted, as well as details regarding their progress. The initial coding procedure was completed by the main author and confirmed with the researcher team when all the codes were discussed until agreement was achieved. This process enhanced the rigour of the findings and improved their credibility and trustworthiness and is a further strength of the study. However, a number of limitations need to be considered in the current study. Firstly, this is a relatively small sample of NGRNs, and therefore, their views and experiences may not be fully reflective of the wider population of national NGRNs in Kuwait. The possible loss of meaning during translation from Arabic to English may also be another limitation of this study.

## 5. Conclusions

The findings of this study provide in-depth insights into Kuwaiti national NGRNs’ experiences, the challenges they encounter, and the reasons behind their decision to stay or leave the nursing profession. Work stress was largely related to excessive workloads, fear of making medical errors, unsupportive and unhelpful colleagues, lack of formal hospital policies, regular staff shortages and low financial remuneration, thereby significantly increasing NGRN turnover in Kuwait. The support strategies reported by Kuwaiti NGRNs in healthcare settings could help to address future NGRNs retention. This study identified the significant differences between the goals of nursing education and the nursing services within the Kuwaiti health service, which resulted in reality shock for NGRNs, and in turn, may affect their retention. Furthermore, this study provides further evidence that Kuwaiti NGRNs’ experiences are largely affected by the negative image and lack of status with regard to nursing as a profession within the Kuwaiti health care service and the wider society. Lastly, there is a need for further studies in this area of concern and for establishing and developing a Kuwait- and Gulf-wide research networks to enable cross-state collaborations and comparisons in relation to nursing workforce needs and developments, including those of NGRNs as they transition into their new nursing roles.

## Figures and Tables

**Table 1 healthcare-10-01856-t001:** Demographic and employment characteristics of NGRNs.

No	Participant ID	Age	Education Level	Months Employed as RN	Work Setting	Length of Orientation	Marital Status
**1**	Khadijah	34	Associate Degree in Nursing	20	Primary Health Care Centre	6 months	Married
**2**	Aisha	21	Associate Degree in Nursing	35	Operating Theatre	9 months	Single
**3**	Asma	23	Certificate	31	Outpatient clinic	9 months	Married
**4**	Hafsa	21	Certificate	15	Surgical word	9 months	Single
**5**	Sarah	32	Certificate	25	Primary Health Care Centre	6 months	Married
**6**	Norah	20	Certificate	25	Primary Health Care Centre	6 months	Single
**7**	Zainab	22	Certificate	25	Primary Health Care Centre	6 months	Married
**8**	Safia	31	Certificate	20	Primary Health Care Centres	6 months	Married
**9**	Mariam	21	Associate Degree in Nursing	18	Medical ward	9 months	Single
**10**	Sheikha	25	Bachelor of Nursing	27	EGD	9 months	Married
**11**	Reem	26	Associate Degree in Nursing	21	Paediatric ward	9 months	Married
**12**	Amina	25	Associate Degree in Nursing	35	Paediatric ward	9 months	Single

## Data Availability

Not applicable.

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
