# Peer review of "The Experience of the Transition from a Student Nurse to a Registered Nurse of Kuwaiti Newly Graduated Registered Nurses: A Qualitative Study"

_healthcare, 2022, doi:10.3390/healthcare10101856_

Round 1
Reviewer 1 Report
Review article titled “The experience of transition from student nurse to registered nurse of Kuwaiti newly graduated registered nurse: A qualitative study”, Healthcare.
1. The study discusses a very important topic, the asymmetric distribution of added value, with global relevance. The phenomena studied is in line with current discussions about the “Great Resignation” and “Quiet Quitting”.
2. The workplace concerns discussed seem to be common in many countries. While they are reasonable and legitimate, some integration may lead to more realistic discussions. For example, advising development of a BSc in Nursing, while per se quite useful, it seems not attractive for students/nurse under the current conditions since at the current educational level the workplace still needs to offer much support. In a similar way, while increasing English proficiency is a necessity for good practice, the value (in real monetary payments) and appreciation for speaking Arabic is not there. Similarly, changing the image of nurses while very much needed seems quite difficult because it is characteristic of society and of power relations (nurse-patient, nurse-medical doctors). The use of nurses from other countries complicates these issues.
3. All recommendations concerning nursing policy, practice, and education seem evident. For example, attracting and retaining nurses, require decent working conditions, including reciprocity of work vis a vis compensation, respect and high-quality relationships in the workplace, and future career development opportunities. I do believe that policy makers know that, but they continue to perhaps take advantage of asymmetric power relations. I mention this because this has to do with the real possibilities, or lack of, change.
4. As you pointed out in the limitations section, a larger and more diverse sample may produce a stronger study. However, given the issues discussed, probably that may not be required. The issues are repetitive, clear, and everywhere.
5. On lines 432-433 you wrote “Secondly, the current study is the aggregation of data across the health regions from urban and rural areas in Kuwait”. This may not be fully supported by the sample size.
Author Response
Reviewer 1 |
|
Reviewer comments |
Authors response |
Line 432-433- reference to the aggregation of data requires review |
Line removed |
Reviewer 2 Report
Revision of the article:
The Experience of Transition from Student Nurse to Registered Nurse of Kuwaiti Newly Graduated Registered Nurses: A qualitative Study
This qualitative article aims to explore the views and experiences of the transition from student nurse to Registered Nurse of national Kuwaiti NGRNs in their first post in clinical practice. The introduction develops a good conceptual framework with relevant bibliography and helps to justify the purpose of the study. In addition, it is methodologically well designed and provides very interesting and relevant results and implications for nursing policy, practice, education and research.
However, with a view to publication, it is necessary to carry out the following areas of improvement:
- - On line 39, instead of [4], [5] should appear [4,5].
- - Reference 12 is missing.
- - It would be appreciated if the inclusion and exclusion criteria were better explained in the recruitment of study participants.
- - In the section on ethical considerations, it would be important to include that the recommendations of the Declaration of Helsinki were followed.
- - It would also be important to explain why the sample is made up entirely of women and not a single man was recruited. This can lead to a gender bias.
- - The title of table 1 appears on the page before the table. Please correct this error in the format.
- - In the introduction section, it would be good to explain the differences between the different studies of the study participants, as well as the clinical practices carried out during the different curricula of the different educational programs.
- - In the first category, taking into account that the family's social support is also discussed, it would be good to broaden the title to for example "nursing and family support". Another possibility is to build another category that encompasses the social support of the family.
- - As stated by the authors in the limitations of the study, it would have been important to expand the sample to obtain more representative and generalizable results.
- - In the contributions of the authors it would be important to explain which author acted as the second evaluator.
- - References are not well referenced according to the regulations of this journal and should be reviewed. You can check the recommendations in: https://www.mdpi.com/journal/healthcare/instructions
We hope these recommendations and suggestions help you to improve the article.
Kind regards
Author Response
Reviewer 2 |
|
Correction of reference order required |
Order corrected -Line 39 |
Reference 12 missing |
Reference added - Line 57 |
Inclusion and exclusion criteria require better explanation |
Inclusion and exclusion criteria reviewed and revised- Lines 165-171 |
In Ethics- include that the recommendations of the Declaration of Helsinki were followed. |
This detail has been added – Lines 174 |
Explain why the sample is made up entirely of women |
Detailed explanation provided - Line 64-87
|
Table 1 title correction required |
Title of table 1 corrected- Line 226 |
Introduction section- explain the differences between the different in studies and clinical practices of the different educational programmes required |
Detailed explanation provided- Lines 64-87
|
Family's social support is also discussed, consider broadening title |
Retitled to Nursing and family support – Line 228 |
Explain which author acted as the second evaluator |
Detail added – Line 565 |
References should be reviewed as per the journal recommendations |
All references are corrected throughout the paper and adhere to the journal recommendations |

Reviewer 3 Report
The background did not include some of the relevant references.
It needs further explanation on the type of qualitative design, whether it is based on the theory, phenomethodological, etc.; this is one of the critical points to improving your work.
In addition, the sampling of the study was not described enough; how did you choose only 12 nurses? Is it enough?
The ethical considerations of your study need to explain more.
The truthiness of the study was not explained.
There is plagiarism in the introduction section; please rewrite the plagiarism sentences in your word.

Author Response
Reviewer 3 |
|
The background did not include some of the relevant references. |
References reviewed and added to the background |
Further explanation on the type of qualitative design used |
Further explanation provided- Line 27-169 |
Sampling requires more description |
More description detailed- Line 165 - 170 |
Ethical considerations require further explanation |
Ethical considerations reviewed and more detail provided- Line 178-180 |
The truthiness of the study was not explained |
Explanation provided- Line 183-203 |
Plagiarism in the introduction section |
Introduction rewritten |

Round 2
Reviewer 3 Report
Dear all
Greeting,
I think the document now is better; however, there are a few important comments that need to be considered:
1- Introduction: background
Although the introduction is better now, most of the sentences added to the introduction focus on general nursing conditions, such as nursing turnover and nursing shortage, either worldwide or in Kuwait, which need to be summarized. In the same line, you need to add a few paragraphs discussing why your topic is significant and a few studies talking about your topic, Transition from Student Nurse to Registered Nurse...
2- Research design
your research design is better now; however, you need to summarise your addition.
3- sampling
The comment in the sampling section has been addressed appropriately.
4- Ensuring rigour
The comment in the rigour section has been addressed appropriately.
Thanks
Author Response
Reviewer comments |
|
1- Introduction: background Summary of nursing turnover and nursing shortage, worldwide and in Kuwait provided and, paragraph added discussing why the topic is significant with supporting studies.
|
Paragraphs discussing why the topic is significant and supporting studies of the transition from Student Nurse to Registered Nurse provided have been added Lines 96-109
|
2- Research design Research design summary required.
|
Research design summary provided - Line 174-175 |
